# A Machine Learning and Explainable AI Framework for Detecting Preclinical Alzheimer's Disease from Naturalistic Driving Behavior

Sai Santosh Reddy Danda*, Alper Kursat Uysal*†, Yi Lu Murphey*,
Amanda Maher‡§, Savannah Rose‡, Carol Persad‡, Robert Koeppe¶, Bruno Giordani‡§
*Department of Electrical, Electronics and Communication Engineering,
University of Michigan-Dearborn, Dearborn, Michigan, 48128, USA
Email: {dssreddy, ukalper, yilu}@umich.edu
†Department of Computer Engineering, Alanya Alaaddin Keykubat University, Alanya, Turkiye
‡Department of Psychiatry, University of Michigan, Ann Arbor, Michigan, 48109, USA
Email: {amhco, savrose, cpersad, giordani}@med.umich.edu
§Michigan Alzheimer's Disease Research Center, Ann Arbor, Michigan, 48109, USA
¶Department of Radiology, University of Michigan, Ann Arbor, Michigan, 48109, USA
Email: koeppe@med.umich.edu

*Abstract*—Alzheimer's disease (AD) is a progressive neurological disorder that primarily affects older adults, with amyloid-beta accumulation serving as a key biomarker of early pathological change. Timely detection of amyloid positivity is critical for enabling early interventions and improving clinical outcomes. However, current diagnostic tools such as PET imaging or CSF analysis are expensive, invasive, and not scalable for widespread screening. In this paper we present TADEC (Trip And Day Level Explainable Classification), a novel data-driven classification framework for automatically identifying amyloid-positive and negative individuals based on their naturalistic driving behaviors. TADEC employs two levels of behavioral modeling Trip-Level, based on individual driving trips, and Day-Level, which aggregates driving behaviors among all the trips taken in the same day, and an explainable AI (XAI) component for generating explanations of prediction results. A comprehensive set of vehicular features categorized into acceleration and jerk patterns, speed behavior, and turn dynamics were extracted. The algorithms in TADEC were implemented, trained, and evaluated using a data collected from 30 amyloid-positive and 35 amyloid-negative consensus-diagnosed cognitively normal older adults. The Trip-Level model achieved the highest classification accuracy of 89.23%, while the Day-Level model reached 86.15%, indicating strong predictive performance and demonstrating that both fine-grained and broader behavioral patterns contribute effectively to classification performances. For the Day-Level approach, average lateral and longitudinal acceleration emerged as the most impactful predictors, whereas trip distance and the number of large negative jerk events were most influential in the Trip-Level approach. TADEC offers a promising, non-invasive, and scalable tool for early detection of preclinical Alzheimer's disease using naturalistic driving behavior.

*Index Terms*—Alzheimer's disease, amyloid status, naturalistic driving, Bayesian hyperparameter optimization, explainable artificial intelligence.

## I. Introduction

As the aging population in the United States continues to grow, the prevalence of dementia is anticipated to rise correspondingly. By 2050, the number of Americans aged 65 and older are expected to increase from approximately 58 million in 2022 to nearly 82 million [1]. Among the various forms of dementia, Alzheimer's disease (AD) is the most prevalent, and represents a significant global public health concern. AD is a progressive neurodegenerative disorder that first presents with impairments in memory, gradually extending to impact behavior, language, visuospatial abilities, and motor coordination, and ultimately death. Projections suggest that AD could affect up to 13.8 million individuals by 2060 [1], highlighting the growing urgency for effective diagnostic and intervention strategies.

A defining neuropathological feature of AD is the accumulation of amyloid-beta ($A\beta$) plaques in the brain [1]. Individuals are classified as amyloid-positive ($A\beta$+) if they exhibit significant cerebral amyloid plaque accumulation, whereas those without such deposits are categorized as amyloid-negative ($A\beta$-). These amyloid plaques can begin to accumulate 15 to 20 years before the clinical symptoms of AD manifest. Current methods for detecting amyloid burden, such as positron emission tomography (PET) imaging and cerebrospinal fluid (CSF) analysis, are invasive, expensive, and typically limited to specialized clinical facilities in well-resourced urban centers. Blood-based biomarkers have not yet been consistently validated, and require laboratory services. As a result, large-scale screening and early identification of at-risk individuals remain challenging. This has created a critical need for alternative, non-invasive, and scalable approaches that can facilitate broader, more accessible early detection of AD-related pathology. Subtle impairments in instrumental activities of daily living (IADLs), such as managing finances, medication adherence, and navigating environments may serve as early indicators of neurodegenerative disease risk and may precede noticeable cognitive decline [2]. These subtle

functional deficits may go undetected by traditional neuropsychological assessments [3][4], yet they are strongly associated with underlying cognitive deterioration [5][6]. As such, sensitive evaluation of complex IADLs presents a valuable, non-invasive avenue for identifying individuals at elevated risk for dementia [7]. Driving is a highly complex instrumental activity of daily living (IADL) that requires continuous coordination of cognitive domains such as spatial awareness, working memory, attentional control, and higher-order executive processes. Early cognitive changes in AD and other neurodegenerative conditions may impair critical driving-related abilities [8], such as situational awareness, visual processing, and response time, even before broader cognitive symptoms become clinically evident [9][10]. These deficits may manifest more prominently in cognitively taxing scenarios, such as turning [11] or navigating complex intersections [12], where rapid judgment and adaptive decision-making are essential. Given the widespread reliance on driving for independence among older adults, evaluation of driving performance offers a promising, ecologically valid, and non-invasive approach for detecting subtle cognitive changes at the earliest stages. In this study, we investigate cognitively normal older adults with elevated amyloid levels, reflecting early accumulation of AD pathology, to understand if they exhibit distinct patterns in naturalistic driving behavior compared to their cognitively normal amyloid-negative counterparts. Using vehicle-based metrics recorded from trips over a one-month period, we aim to uncover subtle differences in driving performance that may serve as early behavioral markers of AD risk, even among those with normal cognition.

Supporting this perspective, several naturalistic driving studies have examined how changes in everyday driving behavior reflect cognitive decline, including early impairment. For instance, Eby et al. [13] found that drivers with early-stage dementia showed restricted patterns and got lost more often, despite similar safety metrics. Roe et al. [14] tracked driving behavior over 2.5 years and found that individuals with preclinical AD took fewer trips, drove shorter distances, and increasingly relied on others. In a year-long naturalistic driving study, Babulal et al. [15] observed that older adults with preclinical AD drove less, avoided night driving, and showed fewer aggressive behaviors. Howcroft et al. [16] used GPS data to examine trip complexity and destination patterns and reported that metrics like mean trip distance helped distinguish cognitive decline. Bayat et al. [17] showed that GPS-derived features effectively differentiated mobility patterns in cognitively intact versus demented individuals. Doherty et al. [18] found that preclinical AD was associated with increased adverse driving behaviors, suggesting driving could serve as an early behavioral marker. Di et al. [19] used random forests to predict mild cognitive impairment (MCI) and dementia. Driving-only models achieved an F1 score of 66%, and 88% when combined with demographics. A custom classifier later reached 96% accuracy [20].

While AI and ML have shown promise in detecting cognitive impairment from driving data, the adoption of explainable artificial intelligence (XAI) [21] remains limited. Most existing studies employ black-box models such as artificial neural networks or random forests, which offer limited interpretability. Additionally, there is a need for intelligent systems that incorporate naturalistic driving behavior across different time granularities (e.g., trip-level vs. day-level) to improve prediction accuracy and generalizability. Addressing these gaps, our study employs interpretable ensemble classifiers and SHAP-based explanation techniques to model amyloid positivity in a cohort of cognitively normal older adults using naturalistic driving data.

This work introduces a novel machine learning framework; TADEC (Trip And Day Level Explainable Classification) distinguishes amyloid-positive from amyloid-negative individuals using a comprehensive set of naturalistic trip features. We employ XGBoost and Random Forest algorithms, optimized through Bayesian hyperparameter tuning, to enhance classification performance. Notably, we introduce a fine-grained, Trip-Level analytical approach, an innovation not previously reported in the literature, that captures driving behavior at a higher temporal resolution. This granularity enables the detection of transient, context-specific deviations, which may signal early cognitive changes but could be overlooked in aggregated analyses (e.g., Day-Level data). The integration of XAI methods further strengthens the framework by providing interpretability and identifying which features most significantly influence model predictions. Collectively, this approach makes both methodological and practical contributions, advancing efforts toward scalable, real-world monitoring of cognitive health in aging populations.

## II. Methodology

### A. Participants and Data Collection

This study used data from 65 adults aged 65 or older, who are consensus diagnosed, and cognitively normal. Based on PET scans, participants were grouped as amyloid-positive ($A\beta$+, $n$ = 30, mean age = 74.1 years, 17 female, 25 white) or amyloid-negative ($A\beta$-, $n$ =35, mean age = 70.8 years, 20 female, 32 white) with similar demographic background. Amyloid status was determined using centiloid values: $\leq 10$ for $A\beta$- and $\geq 20$ for $A\beta$+. All participants had a valid driver's license and drove at least twice per week.

Participants' naturalistic driving behavior was continuously recorded over a period of 30 days, allowing for the collection of data during real-world, self-directed trips without experimental manipulation. Unlike simulated or controlled environments, naturalistic driving captures spontaneous interactions between drivers, vehicles, and their surroundings, offering higher ecological validity and a more accurate representation of everyday behavior. Critically, this type of data is especially valuable for detecting subtle behavioral changes that may reflect early signs of neurological or cognitive decline patterns that often go unnoticed in structured or artificial settings. Vehicular signals were collected at a sampling frequency of 100 Hz using a high-precision data logger, capturing features such as speed, acceleration, distance traveled, and GPS coordinates.

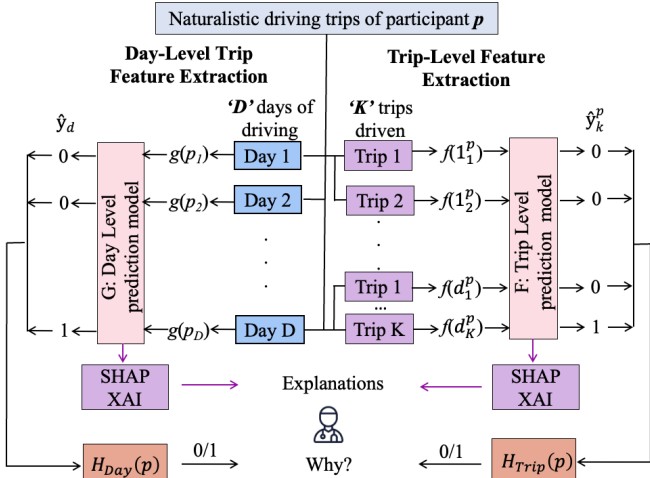

Fig. 1. Overview of the proposed TADEC framework for amyloid positivity classification using Naturalistic Driving Data

Only trips exceeding 0.5 miles in length were included in the analysis to ensure data robustness and relevance. Our study adhered to strict ethical guidelines approved by the institutional review board (IRB). All participants provided informed consent, and data were de-identified prior to analysis.

*B. Trip And Day-Level Explainable Classification (TADEC) Framework for Amyloid Positivity Prediction*

This paper presents a data-driven computational framework, TADEC (Trip And Day-Level Explainable Classification), for identifying amyloid positivity in cognitively normal older adults based on naturalistic driving behavior. TADEC (shown in Figure 1) comprehensively captures individual variability in driving patterns using two modeling strategies using Trip-Level features, and Day-Level trip features, which enables classification systems to learn both fine-grained and composite-level assessments of driving behavioral patterns.

TADEC consists of two major computational stages: (1) classification, which involves building two predictive models, one is trained using Trip-Level features, another using Day-Level trip features, and (2) explainability, which leverages SHAP-based Explainable AI (XAI) method to interpret model predictions and identify the most influential driving features.

The Trip-Level features are extracted from every individual driving trip with the intention of capturing fine-grained behavioral patterns. Let $p \in \mathcal{P}$ represent a participant from the study cohort. Participant $p$ completes a total of $K$ trips across $D$ driving days. Let $d_k^p$ denote the $k^{\text{th}}$ trip taken by the participant $p$ on day $d$, where $k = 1, 2, \ldots, K$ and $d = 1, 2, \ldots, D$. For every trip $d_k^p$, a feature vector $f(d_k^p)$ is extracted, capturing a range of behavioral and contextual driving features, such as average speed, trip distance, etc. These features characterize fine-grained driving behaviors and capture moment-to-moment variability, which may be indicative of subtle differences. A total of 4,195 trips were included in the experiments, where 2,258 trips were recorded from $A\beta+$ participants and 1,937 from $A\beta-$ participants.

The Day-Level trip features are generated from all trips taken in the same day. Let $p_d$ be the set of all the trips taken on the $d^{\text{th}}$ day by participant $p$, the Day-Level trip feature is a vector $g(p_d)$ constructed by the statistics obtains from all the trips in $p_d$. Feature vector $g(p_d)$ encompass spatial and temporal features, and behavioral indicators. More details of these features are presented in Section C. The Day-Level trip features characterize broader behavioral tendencies and mitigates intra-individual variability, potentially enhancing classification stability. A set of 1,167 days of driving trips were analyzed, in which 544 days from $A\beta+$ participants and 623 days from $A\beta-$ participants. A machine learning algorithm was used to train two classification systems: F, a Trip-Level model trained on individual trips, and G, a Day-Level model trained on daily aggregated trip data.

During the prediction process all the feature vectors are sent to their respective predictive models as illustrated in Figure 1. The output from each predictive model is used to generate the final participant prediction result using the Majority Vote functions $H_{\text{Trip}}(p)$ and $H_{\text{Day}}(p)$.

$$H_{\text{Trip}}(p) = \begin{cases} 1, & \frac{1}{K} \sum_{k=1}^{K} \hat{y}_k^p \geq 0.5 \\ 0, & \text{otherwise} \end{cases}$$

$$H_{\text{Day}}(p) = \begin{cases} 1, & \frac{1}{D} \sum_{d=1}^{D} \hat{y}_d \geq 0.5 \\ 0, & \text{otherwise} \end{cases}$$

In the second stage SHAP is used to generate beeswarm plots for explanations that provides visualization of the contribution of individual features to the model's predictions and also the direction of influence on the predicted class.

In cases where the Trip-Level and Day-Level classifiers yield opposite predictions for the same participant, we adopt a human-in-the-loop approach, where domain experts are expected to review both predictions alongside their corresponding SHAP based feature explanations. By comparing short-term (Trip-Level) and aggregated (Day-Level) behavioral patterns, clinicians or researchers can make informed judgments about potential amyloid positivity. This hybrid decision process emphasizes transparency and supports individualized evaluation, especially in borderline or ambiguous cases.

*C. Feature Extraction and Machine Learning Modeling for Amyloid Classification*

For this research, we designed a two-level driving feature extraction approach to capture drivers' habitual patterns, mobility range, risk-related tendencies, and detailed driving dynamics such as speed, acceleration, and jerk. Day-Level features are computed by aggregating driving behaviors across all trips taken in a single day, resulting in 32 features grouped into seven categories. Trip-Level features include 22 measures categorized into five groups that reflect fine-grained driving behavior within individual trips, enabling detection of transient anomalies that may be overlooked in day-level summaries.

TABLE I
SUMMARY OF DAY-LEVEL AND TRIP-LEVEL DRIVING BEHAVIOR
FEATURES

| Feature Category | Day-Level Features | Trip-Level Features |
|---|---|---|
| Acceleration & Jerk Patterns | Average Lateral Acceleration (g) | Average Lateral Acceleration (g) |
| | Average Longitudinal Acceleration (g) | Average Longitudinal Acceleration (g) |
| | Average Jerk | Average Jerk |
| | No. of Large Positive Jerk Events | No. of Large Positive Jerk Events |
| | No. of Large Negative Jerk Events | No. of Large Negative Jerk Events |
| | No. of Deceleration Events > 0.35 g | No. of Deceleration Events > 0.35 g |
| | No. of Deceleration Events > 0.40 g | No. of Deceleration Events > 0.40 g |
| | No. of Deceleration Events > 0.50 g | No. of Deceleration Events > 0.50 g |
| | No. of Deceleration Events > 0.75 g | No. of Deceleration Events > 0.75 g |
| Speed | Average Speed (mph) | Average Speed during Trip (mph) |
| | Maximum Speed of the Day (mph) | Maximum Speed Reached during Trip (mph) |
| | Time Spent Above 60 mph (sec) | Total Seconds Spent Above 60 mph |
| | Time Spent Above 75 mph (sec) | Total Seconds Spent Above 75 mph |
| | Time Spent Above 80 mph (sec) | Total Seconds Spent Above 80 mph |
| Trip Statistics | Average Trip Distance (miles) | Distance of Trip (miles) |
| | Average Trip Duration (minutes) | Duration of Trip (minutes) |
| | Total Number of Trips | |
| | Total Distance Driven (miles) | |
| | Total Driving Time (minutes) | |
| Trip Length Categories | Trips < 5 miles | |
| | Trips Between 5.1–10 miles | |
| | Trips Between 10.1–15 miles | |
| | Trips Between 15.1–20 miles | |
| | Trips > 20.1 miles | |
| Temporal Driving Patterns | Trips During Morning Rush Hour (6–10 am) | Trip Occurred During Morning Rush Hour |
| | Trips During Evening Rush Hour (3–7 pm) | Trip Occurred During Evening Rush Hour |
| | Overnight Trips (12–5 am) | Trip Occurred Overnight (Yes/No) |
| Turning Behavior | Number of Left Turns | Number of Left Turns during Trip |
| | Number of Right Turns | Number of Right Turns during Trip |
| | Left-to-Right Turn Ratio | Ratio of Left to Right Turns |
| Mobility Pattern | Number of Unique Destinations | |
| | Radius of Gyration (miles) | |

This dual-level framework supports both comprehensive behavioral profiling and context-sensitive modeling of naturalistic driving behavior. A detailed description of all extracted features is provided in Table I.

In TADEC framework, two machine learning algorithms, Random Forest (RF) and XGBoost (XGB) were investigated to model the relationship between driving features and amyloid status. RF, an ensemble learning method based on decision trees, was chosen for its robustness in handling high-dimensional data and its ability to reduce variance through bootstrap aggregation. By constructing multiple decision trees and averaging their predictions, RF mitigates overfitting and enhances generalizability, which is particularly important given the complex and noisy nature of naturalistic driving data.

XGB, a gradient boosting framework, was selected for its strong predictive performance and problems involving complex, nonlinear interactions between features. Unlike RF, which builds trees independently, XGB constructs trees sequentially, with each new tree learning to correct the errors of the previous one. This boosting approach results in more accurate and finely tuned models. XGB is also known for its computational efficiency and scalability, making it well-suited to this dataset with a large number of Trip and Day Level records making it a valuable tool in the development of data-driven diagnostic models.

For model training and evaluation, the data were split based on participants rather than by trips or days to make sure the results generalize well to new individuals. Participants' trip data were randomly partitioned into training and testing sets in a stratified way, maintaining the proportion of $A\beta+$ and $A\beta-$ individuals balanced in both sets. Specifically, 70% (21 $A\beta+$, 24 $A\beta-$) of the participants from each group were used for training, 10% (3 $A\beta+$, 4 $A\beta-$) for validation and the remaining 20% (6 $A\beta+$, 7 $A\beta-$) were used for testing.

The model training process begins with the initial training on the training data, followed by a sequence of hyperparameter tuning conducted using the validation set. This allows the model to be fine-tuned and optimized for better generalization before final testing. Once the best set of hyperparameters is determined, the model is retrained using the combined training and validation sets (80% of the data) and evaluated on the test set to assess its final performance.

### D. Hyperparameter tuning via Bayesian Optimization

Bayesian optimization enhances hyperparameter tuning by efficiently navigating the search space using a probabilistic surrogate model. It uses a Gaussian Process (GP) to estimate the objective function while quantifying uncertainty [22]. This approach reduces the number of evaluations required for hyperparameter tuning by progressively refining the search space, aiming to be "less wrong" with each iteration while consistently identifying better-performing hyperparameter configurations. It is particularly advantageous for computationally expensive models.

Bayesian hyperparameter optimization in this study follows a Sequential Model-Based Optimization framework, conducted independently for RF and XGB models. The process begins with a few randomly sampled hyperparameter settings, each evaluated based on model performance. A surrogate model was then constructed to approximate the objective function that maps hyperparameter settings to performance outcomes. Using an acquisition function such as Expected Improvement, the optimizer selects the next promising configuration, effectively balancing the exploration of the search space with the exploitation of high-performing regions. After each evaluation, the surrogate model is updated with the newly obtained results. This iterative process is repeated for up to 30 iterations or until convergence was observed.

### E. Explainable AI using SHapley Additive exPlanations

SHAP (SHapley Additive exPlanations), a state-of-the-art explainable AI (XAI) method grounded in cooperative game theory, was used to quantify the contribution of individual Day-Level and Trip-Level features to classification performance. SHAP assigns a Shapley value to each feature by estimating its average marginal contribution across all possible feature subsets, comparing the model's output with and without each feature under various combinations. By applying the SHAP Python library to the trained RF and XGB models, we systematically interpreted how each feature influenced predictions of $A\beta+$ vs. $A\beta-$. The results were visualized using SHAP beeswarm plots, which depict the distribution of SHAP values for each feature across all observations.

Each row in the beeswarm plot represents a feature, ordered by mean absolute SHAP value (importance). Points are colored by the actual feature value (blue = low, red = high) and positioned along the x-axis based on their SHAP values. Positive SHAP values indicate contribution to the positive class, negative values to the negative class, and values near zero imply minimal effect. This dual encoding allows interpretation of both feature importance and the directional impact of high vs. low feature values on model predictions.

TABLE II
PERFORMANCE COMPARISON OF BASELINE AND OPTIMIZED ML
MODELS ACROSS TRIP-LEVEL AND DAY-LEVEL ANALYSIS

| Metrics | Trip-level Analysis | | | | Day-level Analysis | | | |
| | Baseline | | Optimized | | Baseline | | Optimized | |
| | RF | XGB | RF | XGB | RF | XGB | RF | XGB |
|---|---|---|---|---|---|---|---|---|
| Accuracy | 76.92 | 81.54 | 86.15 | 89.23 | 73.85 | 80.00 | 84.62 | 86.15 |
| Precision | 77.78 | 82.14 | 86.21 | 89.66 | 74.07 | 79.31 | 81.25 | 83.87 |
| Recall | 70.00 | 76.67 | 83.33 | 86.67 | 66.67 | 76.67 | 86.67 | 86.67 |
| F1 Score | 73.68 | 79.29 | 84.75 | 88.13 | 70.18 | 77.93 | 83.82 | 85.25 |

TABLE III
HYPERPARAMETER SPACE FOR RANDOM FOREST ML MODEL

| RF Hyperparameter | Parameter Range | Trip-Level Optimal | Day-Level Optimal |
|---|---|---|---|
| n_estimators | 50–200 | 178 | 156 |
| max_depth | 3–10 | 8 | 9 |
| max_features | sqrt, log2, none | sqrt | sqrt |
| min_samples_leaf | 1–10 | 3 | 3 |
| min_samples_split | 2–10 | 5 | 3 |

TABLE IV
HYPERPARAMETER SPACE FOR XGBOOST ML MODEL

| XGB Hyperparameters | Parameter Range | Trip-Level Optimal | Day-Level Optimal |
|---|---|---|---|
| n_estimators | 50–200 | 144 | 152 |
| max_depth | 3–10 | 9 | 7 |
| learning_rate | 0.01–0.2 | 0.14 | 0.20 |
| colsample_bytree | 0.5–1 | 0.62 | 1.00 |
| gamma | 0–1 | 0.10 | 0.10 |
| alpha | 0–1 | 0.30 | 0.30 |

## III. RESULTS

This section presents a comprehensive analysis of the results obtained from the proposed TADEC framework. Table II presents the performance metrics of the ML models across the two modeling strategies while highlighting the impact of hyperparameter optimization. Tables III and IV detail the ranges of hyperparameters explored during Bayesian optimization. To assess the performance of the classifiers we use four evaluation metrics, Accuracy, Precision, Recall, F1-Score. Accuracy is the percentage of correct predictions. Precision is the proportion of true positives among predicted positives, while recall is the proportion of actual positives correctly identified. The F1 score is the harmonic mean of precision and recall. In this application, the positive class is the $A\beta+$ class and $A\beta-$ is the negative class.

### A. Machine Learning Modeling using Trip-level features

For the Trip-Level classification both RF and XGB classifiers demonstrated strong performances in classifying the participants. The baseline models achieved accuracies of 76.92% and 81.54% for RF and XGB respectively, with XGB outperforming RF across all metrics. After Bayesian hyperparameter optimization, notable improvements were observed in both models. The optimized RF model achieved an accuracy of 86.15%, while the optimized XGB model further improved to 89.23%. The Precision, recall, and F1-score also increased substantially for both models following the optimization. Specifically, the recall improved from 70% to 83.33% for RF and from 76.67% to 86.67% for XGB, indicating enhanced sensitivity to amyloid-positive cases.

### B. Machine Learning Modeling using Day-level trip features

For the Day-Level classification, initially, baseline models achieved moderate classification performance, with RF yielding an accuracy of 73.85% and XGB achieving 80%. After optimization, the accuracy improved to 84.62% for RF and 86.15% for XGB. Similar trends were observed across other performance metrics. Precision increased from 74.07% to 81.25% for RF and from 79.31% to 83.87% for XGB. Notably, recall improved significantly, rising from 66.67% to 86.67% for RF and from 76.67% to 86.67% for XGB. The F1 score reaching 83.82% for RF and 85.25% for XGB post-optimization.

### C. Interpretations of SHAP

To gain deeper insight into the model's decision-making process of classifying amyloid positivity, SHAP values were employed to interpret feature contributions. The SHAP beeswarm plots in Figures 2 and 3 show us that in the Day-Level classification, average longitudinal acceleration, average lateral acceleration, radius of gyration, number of large negative jerk events are the most important features for both RF and XGB ML models. In the Trip-Level classification, number of large negative jerk events, trip distance, average longitudinal acceleration, average lateral acceleration are the most important features for both RF and XGB ML models.

If we take a closer look at the SHAP beeswarm plots for the Day-Level classification framework in Figure 2, radius of gyration exhibited a clear directional pattern: lower values (blue dots) contributed negatively and higher values (red dots) contributed positively to the prediction of amyloid positivity across both models. This suggests that individuals with a broader spatial driving range were more likely to be classified as $A\beta+$, potentially reflecting a tendency toward longer or more dispersed trips. Number of large negative jerk events showed that, higher values (red) pushed predictions toward amyloid-negative, while lower values (blue) pushed predictions toward $A\beta+$. This may indicate that $A\beta+$ individuals tend to brake less suddenly, possibly due to increased caution. Average duration also followed a consistent trend, where shorter durations (blue) were associated with $A\beta-$ predictions and longer durations (red) with $A\beta+$. This may suggest that $A\beta+$ individuals take longer to complete trips. Similarly, left, and right turn counts: fewer turns (blue) were linked with $A\beta-$ predictions, whereas more frequent turning behavior (red) aligned with $A\beta+$. This could reflect differences in route complexity or navigational strategies, $A\beta+$ individuals may be taking routes with more maneuvers due to unfamiliarity or avoidance of high-speed roads.

In the SHAP beeswarm plots corresponding to the Trip-Level classification framework (Figure 3), number of large negative jerk events showed a clear trend, with lower values contributing to amyloid-negative predictions and higher values contributing positively toward $A\beta+$ predictions, indicating that

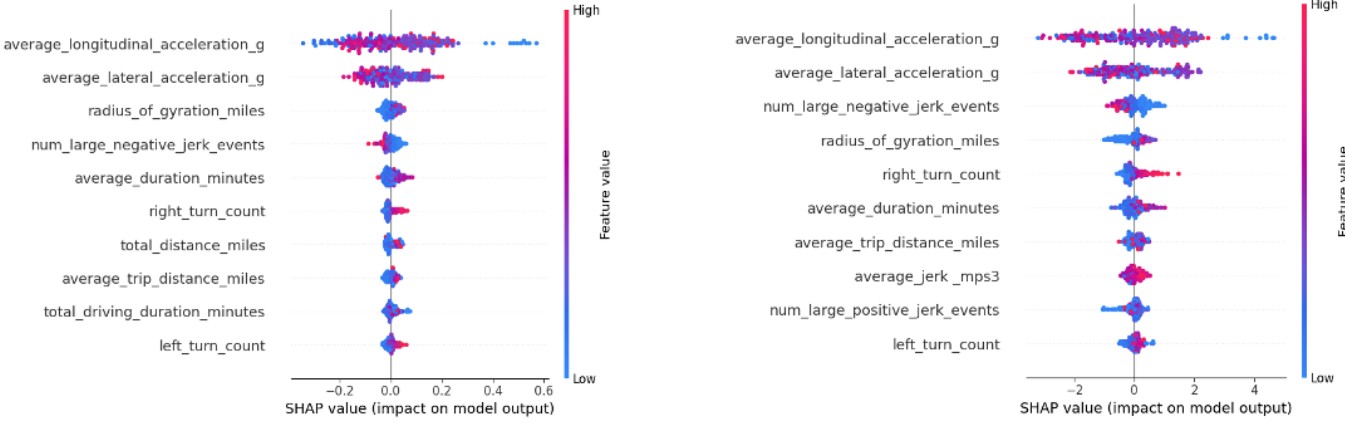

Fig. 2. SHAP Beeswarm Plots for Day-Level Classification Framework (RF-left, XGB-right).

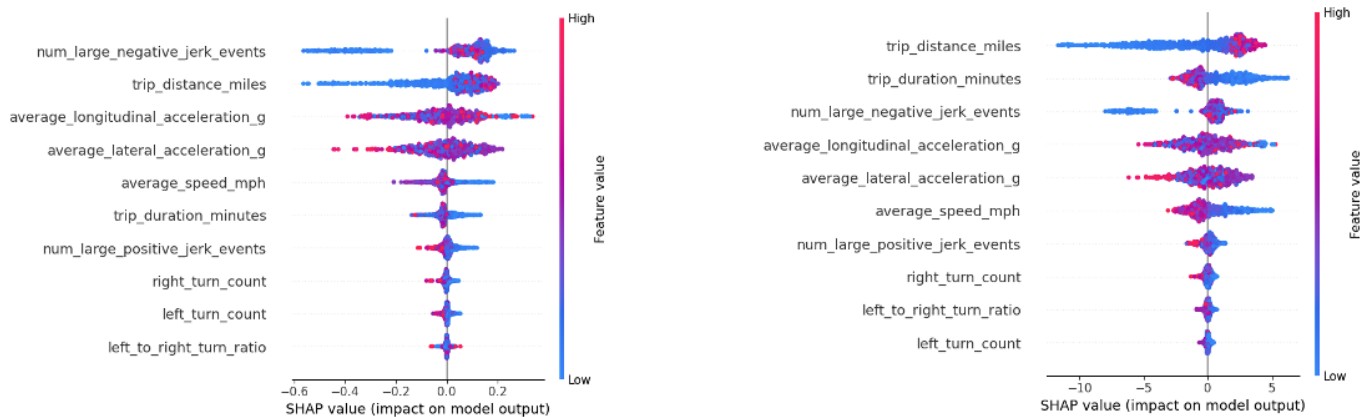

Fig. 3. SHAP Beeswarm Plots for Trip-Level Classification Framework (RF-left, XGB-right).

frequent abrupt braking actions were more characteristic of $A\beta+$ participants. A similar pattern was observed for trip distance, where longer distances were associated with $A\beta+$ participants, and shorter trips aligned with $A\beta-$ participants. For average speed, higher values contributed negatively, while lower values contributed positively to both models. This suggests that $A\beta+$ individuals may be driving at relatively slower speeds, potentially reflecting cautious or conservative driving tendencies. Finally, both left and right turn count demonstrated a directional influence where higher values pushed predictions toward the amyloid-negative class and lower values toward the $A\beta+$ class. This could suggest that $A\beta+$ drivers engage in fewer maneuvering actions during trips, potentially reflecting simpler routes or avoidance of complex intersections and decision-making demands.

While several features were shared between the Trip-Level and Day-Level classification frameworks, certain behavioral metrics (e.g., turn counts, large negative jerk events) exhibited opposite SHAP value directions. This divergence is not contradictory but instead reflects differences in observational granularity. Trip-Level models capture momentary driving behaviors that may reveal in-the-moment compensatory strategies or challenges. In contrast, Day-Level models aggregate multiple

trips to highlight broader driving patterns and routines. Together, these perspectives offer a complementary view of driving behavior in preclinical Alzheimer's disease, underscoring the value of multi-scale analysis in behavioral monitoring.

## IV. CONCLUSION

This paper presents TADEC, an innovative computational framework for the automatic classification of $A\beta+$ and $A\beta-$ individuals based on naturalistic driving behavior. The core of TADEC is a two-level classification approach that builds predictive models using both Trip-Level and Day-Level driving features. We developed a set of 32 Day-Level features and 22 Trip-Level features to comprehensively characterize driving behavior at different temporal scales. Additionally, TADEC incorporates a SHAP-based XAI module to provide interpretable explanations of the model's predictions.

To compare model performances, we employed two machine learning algorithms, RF and XGB, to learn the relationship between two-level driving features and amyloid positivity. These algorithms, combined with Bayesian hyperparameter optimization, were used to develop classification models at both levels. The final classification output was determined through majority voting across predictions from the Day-Level and Trip-Level classifiers. All the classifiers are trained and

evaluated using a set of naturalistic driving trips collected from 65 adults aged 65 or older who were consensus diagnosed as cognitively normal. From this dataset, we extracted 1,167 days of driving trips for training and evaluating the Day-Level models, and 4,195 individual trips for the Trip-Level classification system.

XGB classifiers demonstrated strong performances in classifying $A\beta+$ from $A\beta-$ participants. At both the Trip-Level and Day-Level classifications, XGB consistently outperformed RF across all evaluation metrics. Notably, XGB combined with Bayesian optimization achieved an accuracy of 89.23% at the Trip-Level, and 86.15% at the Day-Level classification. Bayesian hyperparameter optimization led to substantial performance gains. The optimized RF model improved by more than 9 percentage points in accuracy, while the XGB model obtained an improvement of over 4 percentage points in Trip-Level classification accuracy. SHAP Beeswarm Plots offered valuable insights into the relationship between the driving features and classification decision processes in identifying amyloid positivity. Notably acceleration and jerk pattern were the most prominent features in distinguishing the two groups at both levels of classification.

In conclusion, the research findings presented in this paper suggest that the intelligent system developed using the TADEC framework holds significant promise as a low-cost, non-invasive diagnostic tool for detecting preclinical Alzheimer's disease. In future work, we plan to perform external validation, explore transfer learning and advanced deep learning methodologies, alongside subgroup analyses, human-in-the-loop evaluation to strengthen SHAP interpretation and clinical usability.

This work was supported in part by the NIH/NIA Grant R01AG068338 and P30AG072931.

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
