# OpenReview forum: "A Machine Learning and Explainable AI Framework for Detecting Preclinical Alzheimer’s Disease from Naturalistic Driving Behavior"
_IEEE.org/EMBS/BHI/2025/Conference — BHI 2025_

### Official Review · Reviewer_4vKX · 2025-07-07
**A Machine Learning and Explainable AI Framework for Detecting Preclinical Alzheimer’s Disease from Naturalistic Driving Behavior**

**Confidence:** 2
**Clarity Of Writing:** good
**Clinical Significance:** good
**Methodological Novelty:** great
**Overall Rating:** 7

**Experiments And Results:**

good

**Questions For The Authors:**

1. How might variations in environmental factors such as traffic or weather conditions impact the model's performance? Addressing this could help evaluate the robustness of the model under different real-world scenarios.

2. Have you considered the potential influence of vehicle type or driving technology (e.g., adaptive cruise control) on your results? Clarifying this could improve understanding of potential confounding factors affecting model predictions.

3. Could you discuss the feasibility and challenges of implementing TADEC in a practical, clinical screening environment? This clarification would significantly influence the perceived practicality and clinical relevance of the proposed method.

4. What steps would you recommend for external validation of your findings to ensure broader applicability? Understanding your validation strategy would significantly enhance confidence in the model's generalizability and reliability.

5. page length violation,  4-7 pages

**Strengths:**

Clear and clinically relevant objective addressing the need for scalable, non-invasive Alzheimer's screening methods. Innovative dual-level modeling (Trip-Level and Day-Level) to capture both fine-grained and broader driving patterns. Effective use of Bayesian hyperparameter optimization, enhancing model accuracy significantly. Incorporation of SHAP values provides excellent interpretability, critical for clinical acceptance. Robust validation methodology with a clearly defined and balanced dataset. Strong performance metrics, especially at the Trip-Level with accuracy nearing 90%.

**Summary Of The Paper:**

This paper presents a machine learning and explainable AI (XAI) framework called TADEC for detecting preclinical Alzheimer’s disease through naturalistic driving behaviors. The authors employ a dual-level approach—Trip-Level and Day-Level—using Random Forest and XGBoost models optimized through Bayesian hyperparameter tuning. SHAP values are utilized for interpreting model predictions, highlighting the most impactful driving behavior features. TADEC achieved high accuracy, with 89.23% at the Trip-Level and 86.15% at the Day-Level, demonstrating its potential for non-invasive early detection of amyloid positivity

**Weaknesses:**

Limited sample size (65 participants), potentially restricting generalizability. Lack of longitudinal follow-up data to assess progression from amyloid positivity to clinical Alzheimer’s. Potential confounding factors related to driving behavior (e.g., vehicle type, road conditions) not extensively addressed. Absence of external validation or replication with independent cohorts. Although SHAP analysis provides interpretability, further validation or clinical correlation of these interpretations would strengthen the findings.

---

### Official Review · Reviewer_j9y2 · 2025-07-17
**Review highlights the paper's excellent clarity and experiments, great clinical significance and methodological novelty, while noting a limited sample size as potential weaknesses**

**Confidence:** 4
**Clarity Of Writing:** excellent
**Clinical Significance:** excellent
**Methodological Novelty:** excellent
**Overall Rating:** 7

**Experiments And Results:**

great

**Questions For The Authors:**

While the results are strong for this specific cohort, how confident are the authors that these findings will generalize to a broader population of older adults, including those with varying driving habits, different demographic backgrounds, or potentially earlier stages of preclinical AD that might manifest even more subtly?

The authors can provide strong arguments or preliminary data (even if not in the paper) supporting the generalizability or discuss how the framework is designed to adapt to more diverse cohorts (e.g., strategies for data augmentation, transfer learning).

**Strengths:**

a.	Novel framework (TADEC)
b.	Two levels of behavioral modeling
c.	Explainable XAI
d.	Non-Invasive and scalable
e.     Good performance by machine learning models

**Summary Of The Paper:**

This article introduces TADEC, a machine learning framework that uses naturalistic driving behavior to detect preclinical Alzheimer's disease in cognitively normal older adults. It employs both Trip-Level and Day-Level analyses, extracting detailed vehicular features and leveraging explainable AI (XAI) for transparent predictions. The framework demonstrated strong predictive performance, with the optimized XGBoost model achieving 89.23% accuracy at the Trip-Level and 86.15% at the Day-Level, identifying key driving features that differentiate amyloid-positive individuals.

**Weaknesses:**

a.     The study's sample size is still quite small for machine learning models, which could impact generalizability to a wider population.
b.	The article does not mention external validation of the TADEC framework on an independent dataset.
c.	While the abstract mentions that current diagnostic tools (PET imaging,etc) are expensive, invasive, and not scalable, the paper doesn't explicitly compare TADEC's performance or use directly against these established clinical methods.

---

### Official Review · Reviewer_rmfN · 2025-07-22
**A Machine Learning and Explainable AI Framework for Detecting Preclinical Alzheimer’s Disease from Naturalistic Driving Behavior**

**Confidence:** 3
**Clarity Of Writing:** good
**Clinical Significance:** good
**Methodological Novelty:** great
**Overall Rating:** 7

**Experiments And Results:**

great

**Questions For The Authors:**

1. Have you considered how the model’s performance might shift over time? Would a model trained on 30 days of driving generalize data collected over longer periods or across seasonal/weather variations?
2. In instances where Trip-Level and Day-Level classifiers disagree, the framework defers to human review. Could you provide criteria or examples of how clinicians are expected to resolve such cases?
3. Did you perform any feature selection or dimensionality reduction (e.g., PCA, correlation pruning) before classification?
4. Was informed consent obtained regarding continuous behavioral monitoring? And were participants aware their driving behavior would be linked to neurobiological outcomes?

**Strengths:**

- This study addresses a critical public health challenge by focusing on early detection of Alzheimer’s Disease using passive, real-world monitoring.
- The novelty lies in the dual-resolution modeling approach that analyzes driving behavior at both the trip and day levels for richer behavioral insights.
- I also found that feature engineering is carefully organized into meaningful categories (e.g., jerk patterns, temporal driving behaviors, etc.), improving both interpretability and relevance for clinical translation.
- The use of a human-in-the-loop decision strategy, invoking expert review when Trip-Level and Day-Level predictions conflict, reflects thoughtful design that balances automation with clinical oversight.

**Summary Of The Paper:**

This paper proposes TADEC (Trip and Day-Level Explainable Classification), a two-stage machine learning and explainable AI framework for identifying preclinical Alzheimer’s Disease (AD) based on naturalistic driving behavior. The authors trained two models, one at the Trip-Level and another at the Day-Level, using vehicle telemetry data collected over 30 days from 65 cognitively normal older adults, half of whom were amyloid-positive (Aβ+). The classifiers (XGBoost and Random Forest) were optimized using Bayesian hyperparameter tuning. SHAP values were used to interpret model decisions. Results showed high predictive performance (up to 89.23% accuracy), and SHAP analysis revealed that features like jerk, acceleration, and trip distance contributed most to model decisions. TADEC is proposed as a non-invasive, scalable screening tool for early AD risk detection.

**Weaknesses:**

- Although participants are said to be demographically similar, no analysis is presented by age, gender, or education. It would be interesting to see whether model performance varies across these subgroups and whether fairness was evaluated.
- The paper would be strengthened by a discussion of ethical considerations related to passive monitoring for preclinical disease detection, including implications for consent, privacy, and potential misuse of predictions.

---

### Official Review · Reviewer_Jith · 2025-07-22
**A Machine Learning and Explainable AI Framework for Detecting Preclinical Alzheimer’s Disease from Naturalistic Driving Behavior**

**Confidence:** 3
**Clarity Of Writing:** fair
**Clinical Significance:** good
**Methodological Novelty:** good
**Overall Rating:** 6

**Experiments And Results:**

fair

**Questions For The Authors:**

Some key questions remain. Did the authors ensure strict subject-level separation in all training, validation, and test sets to avoid data leakage? How does the model perform across subgroups (e.g., gender, driving frequency, or comorbidities)? Has there been any effort to validate the model using an independent or multi-site dataset? What ethical frameworks are in place to protect the privacy of naturalistic driving data, and how would this system be used in practice with clinicians or caregivers? The proposed human-in-the-loop system involving SHAP explanations is intriguing, but would benefit from an initial usability study or feedback session with end users such as neurologists or geriatricians.

In terms of methodological novelty, the paper makes a strong contribution by introducing a hierarchical behavioral modeling framework for Alzheimer’s detection, applying SHAP to both levels of driving behavior, and utilizing Bayesian optimization for robust model development. These choices elevate the work above many prior black-box ML applications in the space. Overall, this is a timely and relevant study that bridges machine learning, digital health, and neurodegenerative disease diagnostics. With a few improvements—particularly external validation, participant diversity, and human-centered evaluation of explainability—the TADEC framework could be a highly impactful tool for early, non-invasive detection of AD.

**Strengths:**

This work has several notable strengths. First, the two-level modeling strategy is a compelling and innovative way to capture both transient and habitual driving behavior, and is methodologically novel in the context of AD detection. The use of real-world, naturalistic driving data collected longitudinally over a month provides high ecological validity, reflecting everyday function rather than performance in simulated environments. The integration of SHAP-based explainability not only enhances transparency but also provides clinically meaningful insights into which behavioral features distinguish amyloid-positive individuals. Furthermore, the authors use appropriate metrics (accuracy, precision, recall, F1-score), maintain stratified participant-level splits to mitigate data leakage, and employ Bayesian optimization to fine-tune model performance—demonstrating a high level of technical rigor.

**Summary Of The Paper:**

This paper introduces TADEC (Trip And Day-Level Explainable Classification), a novel machine learning framework designed to detect amyloid-beta positivity, an early biomarker for Alzheimer’s Disease (AD), using naturalistic driving behavior from cognitively normal older adults. Using 30 Aβ+ and 35 Aβ− individuals, the authors propose two levels of analysis—Trip-Level and Day-Level—extracting 22 and 32 features respectively, including speed, acceleration, jerk, and turn dynamics. Machine learning models (Random Forest and XGBoost) were trained and optimized using Bayesian hyperparameter tuning. Explainable AI was incorporated using SHAP values. The optimized XGBoost model achieved up to 89.23% accuracy in Trip-Level classification, demonstrating the feasibility of this behavioral biomarker approach.

**Weaknesses:**

Despite its promise, the study has some limitations. The small sample size (n = 65) restricts the statistical power and generalizability of the findings, especially considering potential demographic homogeneity. While the paper mentions consensus cognitive diagnosis and matching, it lacks detail on participant diversity in race, geographic location, or socioeconomic status—factors that may influence driving behavior and model fairness. In terms of model evaluation, the study relies entirely on internal validation; no external dataset or follow-up cohort is used to test the generalizability of the model, which is essential for real-world deployment. Moreover, while SHAP explanations are incorporated, the proposed “human-in-the-loop” decision system remains conceptual—there is no validation or feedback from clinicians to show how these explanations would support decision-making. Finally, although trip-level features boost sample size, the repeated-measures design risks inflating performance unless carefully stratified by subject, which the authors suggest they did, but more methodological transparency would be welcome.